# Hydrothermal Synthesis of Bimetallic (Zn, Co) Co-Doped Tungstate Nanocomposite with Direct Z-Scheme for Enhanced Photodegradation of Xylenol Orange

**Fahad A. Alharthi \*, Wedyan Saud Al-Nafaei, Alanoud Abdullah Alshayiqi, Hamdah S. Alanazi and Imran Hasan \***

Department of Chemistry, College of Science, King Saud University, Riyadh 11451, Saudi Arabia
\* Correspondence: fharthi@ksu.edu.sa (F.A.A.); iabdulateef@ksu.edu.sa (I.H.)

**Abstract:** In the present study, pristine $ZnWO_4$, $CoWO_4$, and mixed metal $Zn_{0.5}Co_{0.5}WO_4$ were synthesized through the hydrothermal process using a Teflon-lined autoclave at 180 °C. The synthesized nanomaterials were characterized by various spectroscopic techniques, such as TEM, FTIR, UV–vis, XRD, and SEM-EDX-mapping to confirm the formation of nanocomposite material. The synthesized materials were explored as photocatalysts for the degradation of xylenol orange (XO) under a visible light source and a comparative study was explored to check the efficiency of the bimetallic co-doped nanocomposite to the pristine metal tungstate NPs. XRD analysis proved that reinforcement of $Co^{2+}$ in $ZnWO_4$ lattice results in a reduction in interplanar distance from 0.203 nm to 0.185 nm, which is reflected in its crystallite size, which reduced from 32 nm to 24 nm. Contraction in crystallite size reflects on the optical properties as the energy bandgap of $ZnWO_4$ reduced from 3.49 eV to 3.33 eV in $Zn_{0.5}Co_{0.5}WO_4$, which is due to the formation of a Z-scheme for charge transfer and enhancement in photocatalytic efficiency. The experimental results suggested that $ZnWO_4$, $CoWO_4$, and $Zn_{0.5}Co_{0.5}WO_4$ NPs achieved a photocatalytic efficiency of 97.89%, 98.10%, and 98.77% towards XO in 120 min of visible solar light irradiation. The kinetics of photodegradation was best explained by pseudo-first-order kinetics and the values of apparent rate const ($k_{app}$) also supported the enhanced photocatalytic efficiency of mixed metal $Zn_{0.5}Co_{0.5}WO_4$ NPs towards XO degradation.

**Keywords:** nanocomposites; bimetallic Z-scheme; photoreduction; hydrothermal reaction; organic pollutants

## 1. Introduction

The development of societies and industries has led to the deterioration of freshwater qualities through the release of untreated effluents containing hazardous chemical contaminants, such as dyes and pharmaceutical products, etc. [1,2]. The toxic organic compounds, especially dyes and pharmaceuticals, contaminated wastewater in the environment, which can cause a variety of health issues to humans [3,4]. Among various organic pollutants, xylenol orange (XO; $C_{31}H_{32}N_2O_{13}S$) belongs to the acidic dye family commonly used in the textile, paper, and printing industries, but beyond the permissible limit, it causes nervous system breakdown, liver–kidney damage, blood disorders, and skin irritation [5,6]. So, it has become a matter of serious concern to develop new technology-based methods and resolutions which can treat these hazardous chemicals in the water and regulate their limits in the aquatic environment. Among various chemical, physical, and biological wastewater treatment methods, photocatalysis enables the adsorbed organic molecules to be degraded at a low cost into $H_2O$ and $CO_2$ using solar irradiation, which is abundant, renewable, and environmentally friendly [7,8]. Moreover, advanced oxidation processes (AOPs) were created to promote the removal or degradation of contaminant species through redox processes, focused on the remediation of contaminants in wastewater; they are a group of techniques that use reactive oxidant species (ROS) to mineralize a wide range of resistant organics [9,10].

Metal tungstate nanoparticles (NPs) have received a lot of interest among the different types of semiconductor photocatalysts [11]. For this reason, the study and synthesis of single photocatalysts that successfully use visible light to stimulate photocatalysis have recently become a promising idea. Until now, various semiconductors with sufficient energy band gaps to absorb visible light, such as $Fe_2O_3$, $Ag_2O$, $Bi_2WO_6$, g-$C_3N_4$, $MnWO_4$, $CdWO_4$, CdS, $NiMoO_4$, $Cu_2O$, $In_2S_3$, and $Si_3N_4$, have been effectively synthesized and utilized for photocatalysis, because of its high average refractive index, high chemical stability, excellent light absorbing property, and high catalytic activity [12–14]. In the tungstate family, zinc tungstate ($ZnWO_4$) has been a widely known semiconductor photocatalyst owing to its high physical and chemical stability and difficulty to dissolve [15]. However, due to homogeneity, it is only UV light active and possesses a high rate of electron–hole pair recombination, which poses an obstacle in the practical applications of $ZnWO_4$ [16]. Various methods have been applied previously to increase the photocatalytic efficiency of $ZnWO_4$, such as semiconductor coupling [17], noble metal loading [18], and metal deposition [19]. The metal deposition of $ZnWO_4$ with other metals having a narrow band gap could effectively inhibit the electron–hole pair recombination by expanding the light absorption range, and thus a robust enhancement in photocatalytic efficiency will appear [20,21]. So, the Co deposition method was applied in this study to synthesize the Co-doped nanocomposite $Zn_{0.5}Co_{0.5}WO_4$ with improved energy bandgap, optical properties (UV–vis light active), and photocatalytic efficiency. Since the ionic radii of both $Co^{2+}$ and $Zn^{2+}$ are approximately the same, it can be easily deposited in the solid matrix of $ZnWO_4$ and result in the reduction of the energy bandgap from 3.87 eV ($ZnWO_4$) to 1.95 eV ($Zn_{0.5}Co_{0.5}WO_4$). $ZnWO_4$ was identified as a large band gap semiconductor (band gap = 3.87 eV) with the appropriate conduction band and valance band locations based on its electronic characteristics, while, on the other hand, cobalt tungstate ($CoWO_4$) has been recognized as having excellent phase composition, strong chemical stability, and a low energy bandgap, which makes it adaptable to degrade the dyestuff in the visible light range [22].

In the present work, a hydrothermal method was used for the synthesis of pristine metal tungstate nanomaterials ($ZnWO_4$, $CoWO_4$ NPs) and mixed metal tungstate nanocomposite ($Zn_{0.5}Co_{0.5}WO_4$ NPs) using a Teflon line autoclave. The synthesized NPs were explored for the photocatalytic degradation of XO dye and a comparative study was designed to explore the photocatalytic mechanism and efficiency of the photocatalyst.

## 2. Results and Discussion

### 2.1. Material Characterization

Figure 1 represents the FTIR of $ZnWO_4$, $CoWO_4$, and $Zn_{0.5}Co_{0.5}WO_4$, in which the characteristic peaks of $ZnWO_4$ are 3448 and 1644 $cm^{-1}$ (stretching and bending vibrations of –OH group) due to surface adsorbed water, 473 to 881 $cm^{-1}$ ($W_2O_8$ bonds ($WO_6$) groups shared at edges and $WO_2$ groups), and 473 $cm^{-1}$ (stretching vibrational of the Zn–O) [23]. Three peaks at 541, 685, and 716 $cm^{-1}$ correspond to symmetric and asymmetric stretching vibrations of the bridging atom of the $WO_2$ groups of distorted octahedral ($WO_6$) clusters [24]. Broad absorption bands at 816 $cm^{-1}$ and 881 $cm^{-1}$ are caused by symmetrical vibrations of bridge oxygen atoms of Zn–O–W groups [24]. Similar types of stretching and vibrational bands are also shown by $CoWO_4$ with different peak values and 469 $cm^{-1}$ belongs to the Co–O bond. As compared to $ZnWO_4$, the FTIR of mixed metal $Zn_{0.5}Co_{0.5}WO_4$ showed the difference in peak shapes and intensities due to the partial replacement of $Zn^{2+}$ ions by $Co^{2+}$ ions [25].

The formation of $ZnWO_4$, $CoWO_4$, and $Zn_{0.5}Co_{0.5}WO_4$ was confirmed by the XRD analysis. Figure 2 shows the XRD pattern of the $ZnWO_4$, $CoWO_4$, and $Zn_{0.5}Co_{0.5}WO_4$ prepared by the hydrothermal method at 180°C for 24 h. The XRD spectra of $ZnWO_4$ shows characteristic peaks at 2θ value of 15.58°, 19.04°, 23.94°, 24.66°, 30.59°, 31.38°, 36.54°, 38.47°, 41.41°, 44.40°, 46.55°, 50.39 51.84°, 53.76°, 54.11°, 61.85°, 64.90°, and 68.34°, which belong to miller indices (010), (100), (011), (110), (111), (020), (021), (200), (121), (112), (211), (220) (130), (221), (202), (113), (132), and (041), respectively (JCPDs card no. 96-210-1675). Then,

the XRD spectra of $CoWO_4$ shows characteristic peaks at 2θ value of 19.02°, 23.88°, 27.72°, 31.50°, 36.44°, 38.69°, 48.52°, 52.17°, 61.89°, and 65.14°, which belong to miller indices (100), (011), (110), (020), (002), (200), (022), (031), (310), and (040), respectively (JCPDs card no. 96-591-0317). Finally, the XRD pattern of $Zn_{0.5}Co_{0.5}WO_4$ NPs shows peaks ascribed to the $ZnWO_4$ at 15.54° (010), 36.41° (021), 41.35° (211), 51.95° (220), 53.94° (130), and 68.56° (041), and peaks ascribed to the $CoWO_4$ at 48.81° (022), 61.81° (310), and 64.96° (040), respectively, which suggests that Co has been successfully doped in the solid matrix of $ZnWO_4$. Moreover, the peak intensities of $Zn_{0.5}Co_{0.5}WO_4$ NPs are greater than pristine $ZnWO_4$ suggesting the increase in crystallinity of the complex upon being mixed with $Co^{2+}$, which has been proved by data given in Table 1. There are several extra peaks observed in the XRD spectra of $Zn_{0.5}Co_{0.5}WO_4$ which do not belong to either $ZnWO_4$ or $CoWO_4$. These peaks are due to the formation of $Na_2W_2O_7$ and $Na_2W_4O_{13}$ phases with the $Zn_{0.5}Co_{0.5}WO_4$ phase [26,27].

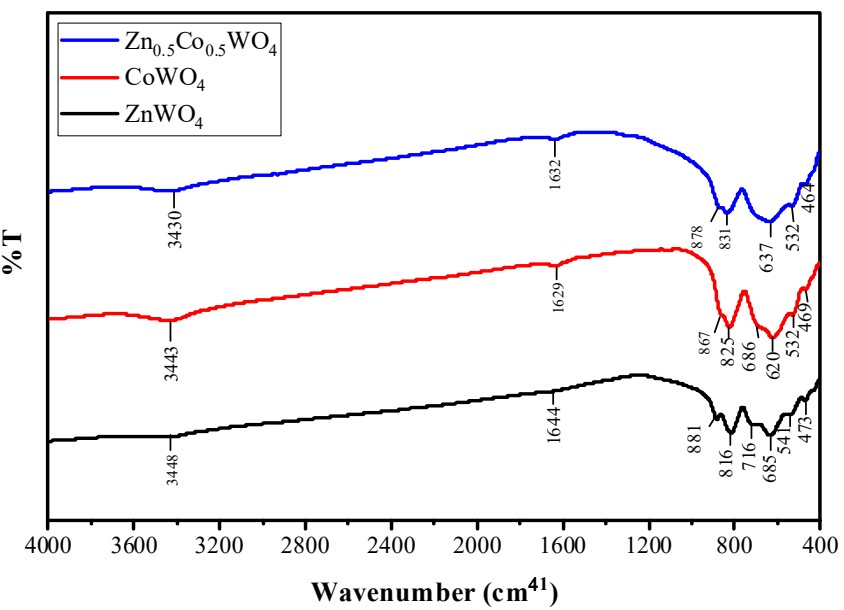

**Figure 1.** FTIR spectra of $ZnWO_4$ (black line), $CoWO_4$ (purple line), and $Zn_{0.5}Co_{0.5}WO_4$ (light blue line).

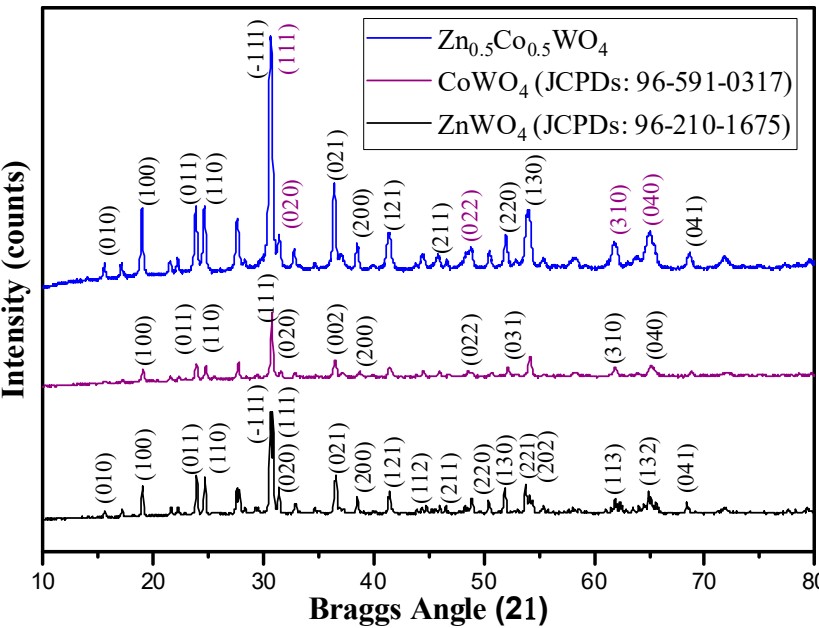

**Figure 2.** X-ray diffraction pattern of $ZnWO_4$, $CoWO_4$, and $Zn_{0.5}Co_{0.5}WO_4$ calcined at 600 °C.

**Table 1.** XRD parameter of ZnWO$_4$, CoWO$_4$, and Zn0.5Co0.5WO$_4$.

| Crystallinity (%) | Dislocation Density (δ) Lines ($10^{14} \times$ m$^2$) | Crystallite Size (nm) at 2θ value | Interlayer Spacing (nm) at 2θ | FWHM ($\beta_{hkl}$) | 2θ | Component |
|---|---|---|---|---|---|---|
| 45.24 | 9.78 | 31.98 | 0.203 | 0.45 | 30.72 | ZnWO$_4$ |
| 48.46 | 9.65 | 32.19 | 0.195 | 0.81 | 30.05 | CoWO$_4$ |
| 42.18 | 17.02 | 24.24 | 0.198 | 0.76 | 30.33 | Zn$_{0.5}$Co$_{0.5}$WO$_4$ |

Further information about crystallite size and dislocation density; Scherrer's equation was taken into consideration [28].

$$D = \frac{0.9\lambda}{\beta \cos\theta} \tag{1}$$

$$\text{Dislocation Density}(\delta) = \frac{1}{D^2} \tag{2}$$

$$\text{Interlayer Spacing}(d_{111}) = \frac{n\lambda}{2 \sin\theta} \tag{3}$$

$$\%\text{Crystallinity} = \frac{\text{Area under the crystalline peaks}}{\text{Total area}} \times 100 \tag{4}$$

where D is the crystallite size, $\lambda$ is the characteristic wavelength of the X-ray, β represents the angular width in radian at an intensity equal to half of its maximum (HWMI) of the peak, and $\theta$ is the diffraction angle. Using Equation (1), the average particle size of ZnWO$_4$, CoWO$_4$, and Zn$_{0.5}$Co$_{0.5}$WO$_4$ was 31.98, 32.19, and 24.24 nm suggesting a contraction in particle size upon Co inclusion in the ZnWO$_4$ solid lattice.

The surface morphology of the ZnWO$_4$, CoWO$_4$, and Zn$_{0.5}$Co$_{0.5}$WO$_4$ were studied using SEM analysis given in (Figure 3). ZnWO$_4$ reveals an aggregation of irregularly shaped crystals with plenty of nanorods (Figure 3a). CoWO$_4$ exhibit uniform nanoparticles of various size (Figure 3b), while Zn$_{0.5}$Co$_{0.5}$WO$_4$ revealed an assembly of fine nanoparticles with a porous surface (Figure 3c). The elemental composition of the synthesized Zn$_{0.5}$Co$_{0.5}$WO$_4$ NPs was observed by EDX analysis and the results showed the presence of the element (weight%) as O (28.52%), Zn (4.55%), Co (4.22%), and W (62.81%), which confirms the successful deposition of Co$^{2+}$ in the ZnWO$_4$ lattice (Figure 3d). The high atomic percentage of W in the EDX spectra is due to the presence of impurity in the form of Na$_2$W$_2$O$_7$ and Na$_2$W$_4$O$_{13}$ phases with the Zn$_{0.5}$Co$_{0.5}$WO$_4$, which resulted in an extra amount of W in the EDX spectra [27].

Furthermore, to figure out the size of the ZnWO$_4$, CoWO$_4$, and Zn$_{0.5}$Co$_{0.5}$WO$_4$, TEM analysis was performed (Figure 4a–c). ZnWO$_4$ (Figure 4a) is composed of particles with a rod-like shape; CoWO$_4$ (Figure 4b) presented a morphologically spherical shaped particle. The synthesized material Zn$_{0.5}$Co$_{0.5}$WO$_4$ (Figure 4c) exhibited particles mixed of nanorods with some spherical particles with size in a range from 5 to 60 nm, and the Gaussian distribution of the average size was found to be 31 nm given in Figure 4d. Figure 4e represents the TEM image of Zn$_{0.5}$Co$_{0.5}$WO$_4$, which suggested the shape of particles to be distorted monoclinic due to the doping of Co into the ZnWO$_4$ lattice. Selected area electron diffraction (SAED) analysis (Figure 4f) was further considered to simulate the TEM data with XRD to assess the purity and shape of the nanoparticles. It contains patterns originating from diffraction points in the TEM screen belonging to randomly oriented nanoparticles. The occurrence of diffraction patterns in Figure 4f suggested that the material is crystalline and pure. The hkl values belonging to yellow fringes simulate the XRD hkl planes of Zn$_{0.5}$Co$_{0.5}$WO$_4$ suggesting the shape of the nanoparticles to be monoclinic.

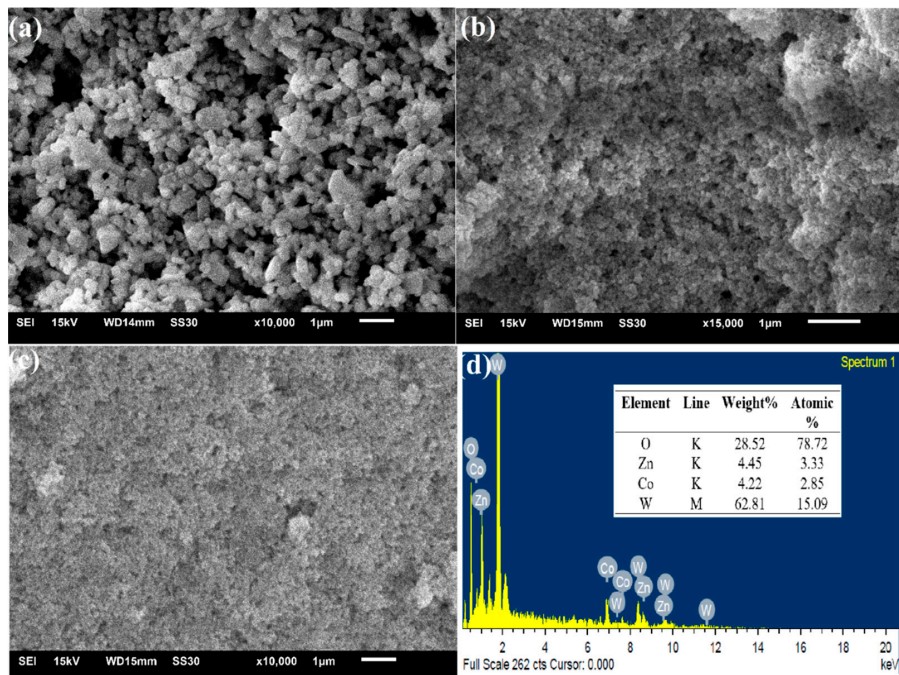

**Figure 3.** SEM micrographs of (**a**) ZnWO$_4$ NPs; (**b**) CoWO$_4$ NPs; (**c**) Zn$_{0.5}$Co$_{0.5}$WO$_4$ at 1 μm scale; (**d**) EDX spectra of Zn$_{0.5}$Co$_{0.5}$WO$_4$ in 0–20 keV range.

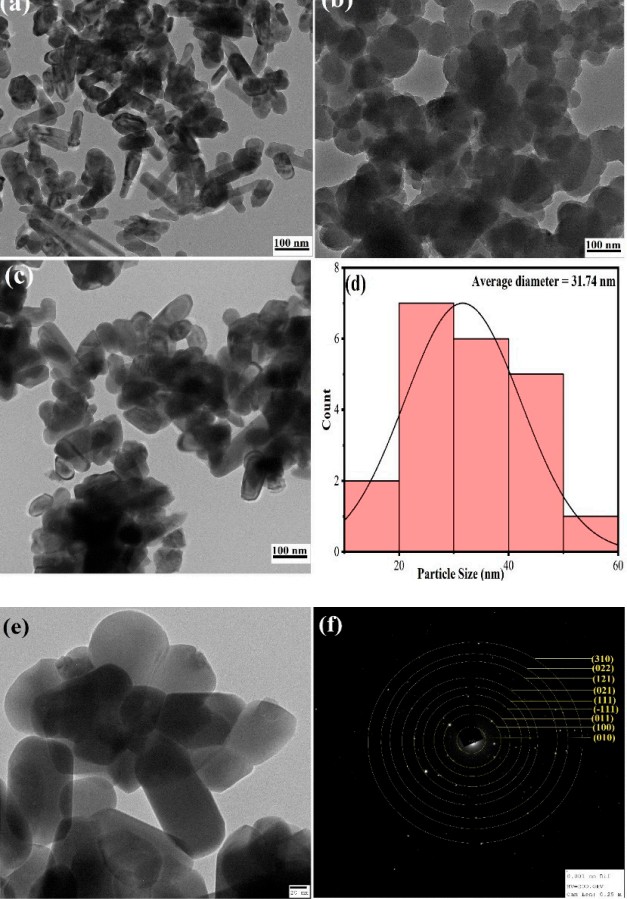

**Figure 4.** TEM images of (**a**) ZnWO$_4$; (**b**) CoWO$_4$; (**c**) Zn$_{0.5}$Co$_{0.5}$WO$_4$ at 100 nm magnification range; (**d**) particle size histogram; (**e**) TEM image of Zn$_{0.5}$Co$_{0.5}$WO$_4$ at 20 nm magnification range; (**f**) selected area electron diffraction (SAED) image of Zn$_{0.5}$Co$_{0.5}$WO$_4$.

The optical properties of the synthesized NPs and their energy bandgap were estimated by using UV–vis spectroscopy in the range of 200–700 nm and the results are given in Figure 5. The absorption spectra of all the synthesized NPs $ZnWO_4$, $CoWO_4$, and $Zn_{0.5}Co_{0.5}WO_4$ exhibited two peaks, one around 260–312 nm and another broad peak around 530–650 nm, suggesting the UV–vis activeness of the NPs. The inset in Figure 5 shows Tauc's plot, which is used to calculate band gap energy (Eg) of the prepared samples by using the Equation (5) [29];

$$(\alpha h\nu) = A(h\nu - E_g)^n \tag{5}$$

where $\alpha$ is the absorption coefficient, A is constant, h is plank constant, $\nu$ is the frequency of radiations, and n is a constant of transition variations, i.e., n = 1/2 for direct transitions and n = 2 for the indirect transitions. Tauc's plot indicated that the synthesized $ZnWO_4$, $CoWO_4$, and $Zn_{0.5}Co_{0.5}WO_4$ were found to have an energy bandgap value of 3.49 eV, 3.42 eV, and 3.33 eV, respectively. Results showed that the deposition of $Co^{2+}$ in the $ZnWO_4$ solid lattice results in the increase in absorption edge by decreasing the energy bandgap, which inhibits the rate of electron–hole pair recombination, and thus improves the photocatalytic efficiency of the synthesized material [30]. The doping of $Co^{2+}$ into $ZnWO_4$ leads to the induction of deformations in the pure monoclinic lattice of $ZnWO_4$, which creates mobile oxygen vacancies in the cluster and makes an improvement in the catalytic [31,32].

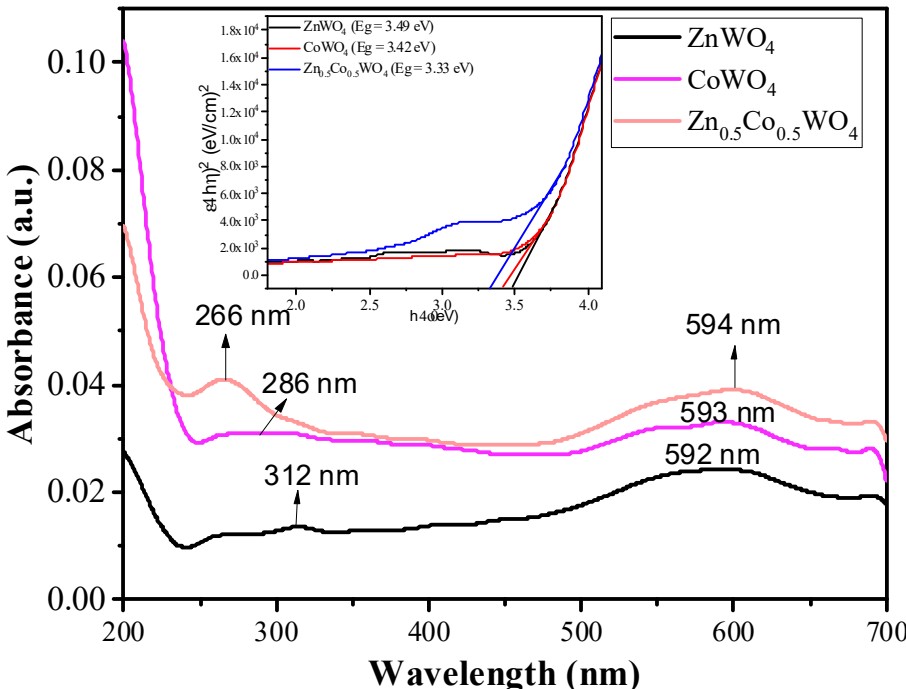

**Figure 5.** UV–vis plot of $ZnWO_4$ NPs, $CoWO_4$, and $Zn_{0.5}Co_{0.5}WO_4$ in the wavelength range of 200–700 nm (inset is the Tauc's plot for calculating the band gap energy ($E_g$) of the material).

Figure 6 consists of the photoluminescence emission spectra of $ZnWO_4$, $CoWO_4$, and $Zn_{0.5}Co_{0.5}WO_4$ recorded at a 594 nm excitation wavelength. It can be seen that the PL spectra spans in the range of 545 to 575 nm with a prominent emission peak at 558 nm. The information obtained from the PL spectra suggested that the PL intensity of synthesized $Zn_{0.5}Co_{0.5}WO_4$ was found to be lower than that of $ZnWO_4$ and $CoWO_4$. The inclusion of Co in the wolframite monoclinic structure of $ZnWO_4$ results in emissions due to the radiative transition between W and O in the $WO_6^{6-}$ molecular complex, which effectively slows down the electron–hole [33–35].

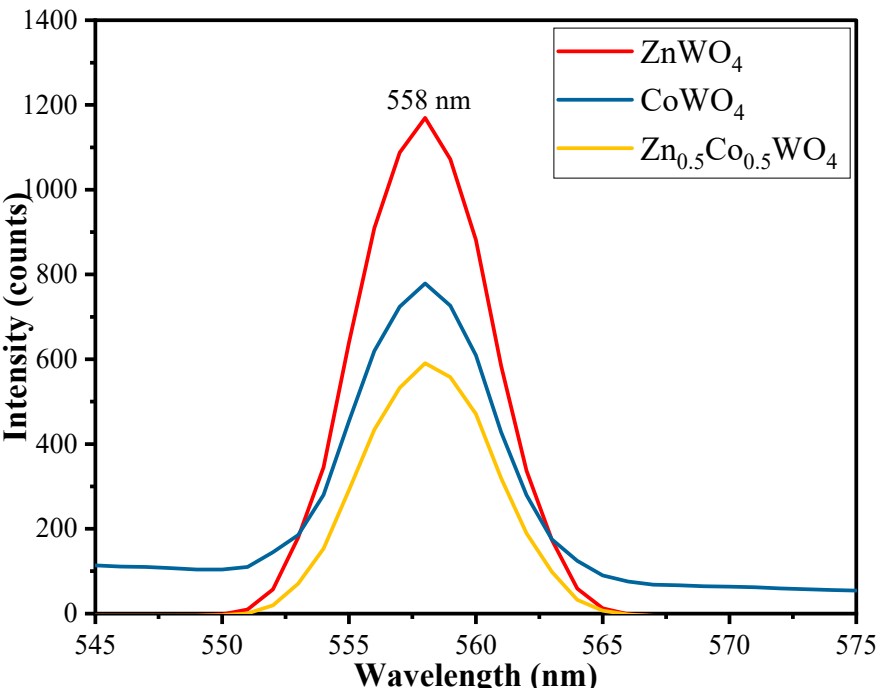

**Figure 6.** Photoluminescence (PL) spectra of $ZnWO_4$, $CoWO_4$ and $Zn_{0.5}Co_{0.5}WO_4$ were recorded at a 594 nm excitation wavelength dispersed in methanol.

## 2.2. Photocatalytic Experiments and Optimization of Reaction Parameters

### 2.2.1. Effect of Variable XO Concentration

The concentration of the dye solution is an important parameter for elucidating the photocatalytic efficiency of the material. Photocatalytic experiments were conducted by taking 10 mg of NPs as a catalyst with variable (20–100 ppm) XO concentration for 120 min of visible light irradiation. The results obtained are given in Figure 7a–d, which suggest that $Zn_{0.5}Co_{0.5}WO_4$ possesses better photocatalytic efficiency (97.17%) as compared to $CoWO_4$ (91.81%), followed by $ZnWO_4$ (89.51%) for 60 ppm of XO, but as the concentration of XO increases beyond 60 ppm, the percentage degradation decreases (Figure 7d). The causes could include: (1) XO may cover more $ZnWO_4$, $CoWO_4$, and $Zn_{0.5}Co_{0.5}WO_4$ active sites, which inhibits the production of oxidants ($^{\bullet}OH$ or $^{\bullet}O_2^{-}$ radicals) and lowers the efficiency of degradation: and (2) higher XO concentration absorbs a greater number of photons, and as a result, there are not enough photons to activate the surfaces of $ZnWO_4$, $CoWO_4$, and $Zn_{0.5}Co_{0.5}WO_4$ nanoparticles, which slows down XO's degradation at higher concentrations [36]. So, 60 ppm XO concentration was chosen as the optimum concentration for further photocatalytic experiments.

### 2.2.2. Effect of Variable Catalyst Dose

The physicochemical characteristics of nanomaterials, such as surface area, phase structure, particle size, and interface charge, make them able to complete adsorption action [27]. To efficiently degrade XO through catalysis using the catalyst, the photocatalytic experiment was performed by varying catalyst doses, such as 5, 10, 15, and 20 mg with 10 mL of 10 ppm XO. The results given in Figure 8a–d suggested that the photocatalytic efficiency increased from 5 mg catalyst dose to 10 mg and starts declining with further increase. All the synthesized NPs exhibit maximum photocatalytic efficiency at 10 mg catalyst dose as 97.89% for $ZnWO_4$, 98.10% for $CoWO_4$, and 98.77% for $Zn_{0.5}Co_{0.5}WO_4$. More active sites are produced on the catalyst's surface at lower concentrations, which promotes the creation of $^{\bullet}OH$ or $^{\bullet}O_2^{-}$ radicals. After that, increasing the catalytic dose causes the particles to agglomerate and sediment, which increases the turbidity and opacity of the slurry. Hence,

the generation of $^\bullet OH$ or $^\bullet O_2{}^-$ radicals automatically decrease [24]. Therefore, the catalyst dose of 10 mg was chosen to be the optimum dose for the degradation.

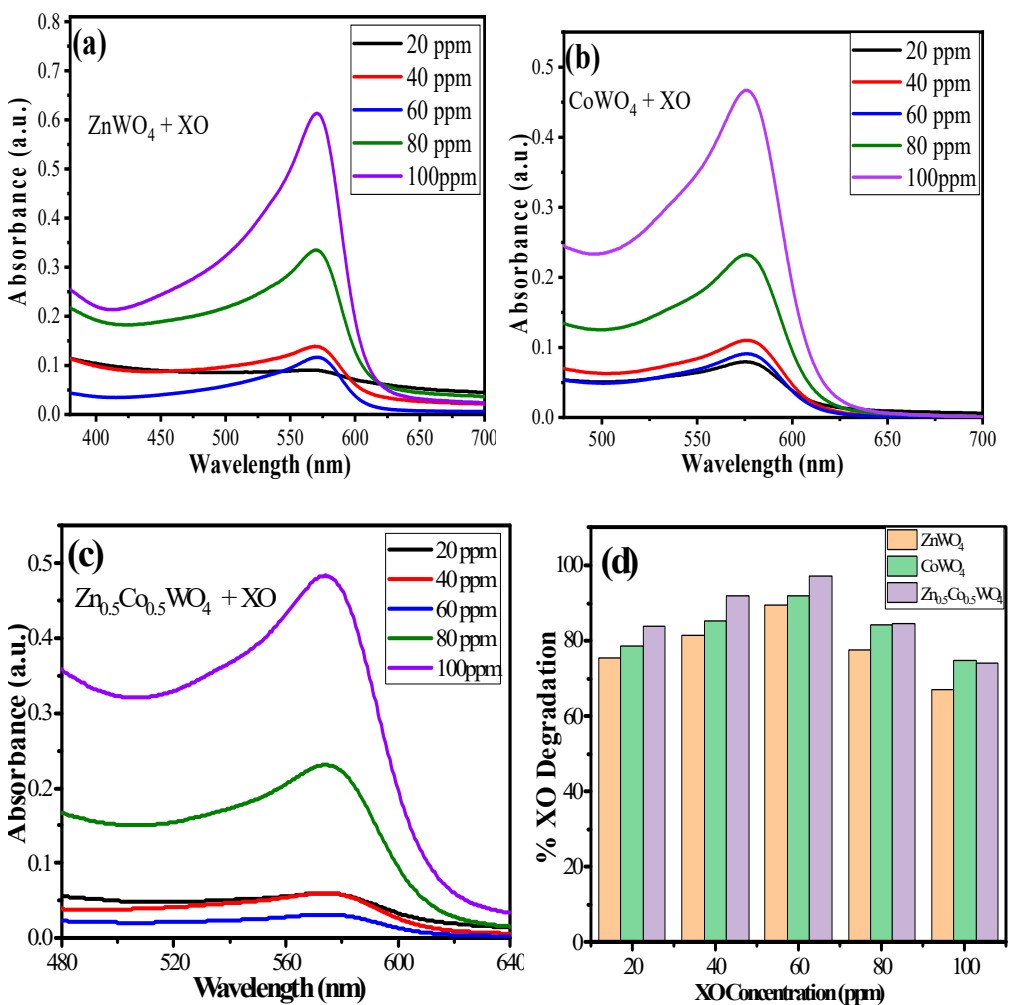

**Figure 7.** Effect of initial concentration of XO dye on photocatalytic degradation by (**a**) ZnWO$_4$; (**b**) CoWO$_4$; (**c**) Zn$_{0.5}$Co$_{0.5}$WO$_4$; (**d**) XO degradation (%) vs. XO concentration bar graph.

### 2.2.3. Effect of pH

Experiments were performed to evaluate the impact of pH on the photocatalytic efficiencies of ZnWO$_4$, CoWO$_4$, and Zn$_{0.5}$Co$_{0.5}$WO$_4$. A total of 10 mg of the catalyst was dispersed in 20 mL of 60 ppm XO solution with a variable pH range from 1–7, and irradiated for 120 min under the visible light source. The obtained results after the experiments are given in Figure 9, which suggests that the photocatalytic efficiency of all the catalysts increased until pH 3, giving a maximum value of 93.35% for ZnWO$_4$, 96.21% for CoWO$_4$, and 97.97% for Zn$_{0.5}$Co$_{0.5}$WO$_4$. Further, an increase in pH value beyond 3 results in a decrease in photocatalytic efficiency with a continuous fall until pH 7. So, pH 3 was taken as the optimum pH value for the photocatalytic experiments. The trend can be explained by the fact that as XO is an anionic dye, it can easily bind with the positive surface of the catalyst at low pH through electrostatic interactions, and thereby, in the presence of light, it gets degraded by $^\bullet OH$ or $^\bullet O_2{}^-$ radicals [7]. As the pH values increase, the surface becomes negative, which will exert a repulsive force on the anionic XO molecule, and hence, the photocatalytic efficiency decreases.

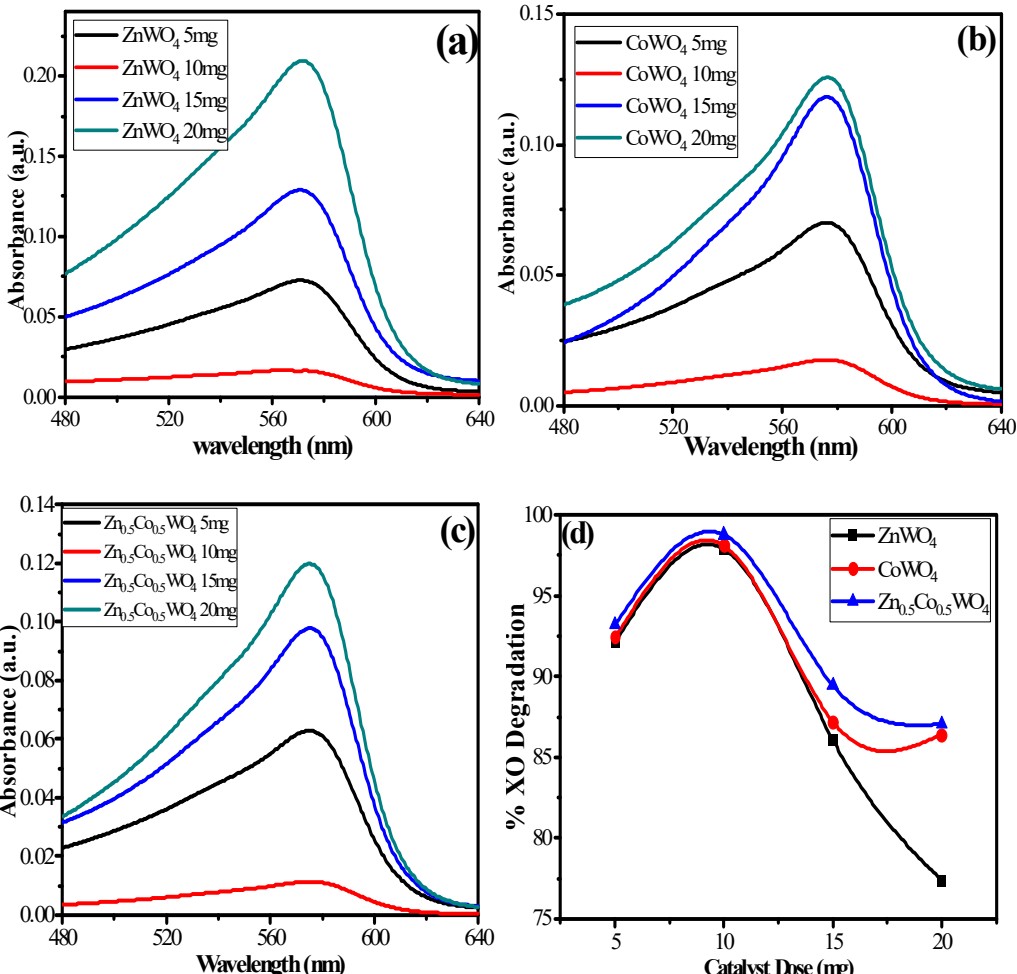

**Figure 8.** Photocatalytic degradation of XO with a different catalytic dose of (**a**) $ZnWO_4$; (**b**) $CoWO_4$; and (**c**) $Zn_{0.5}Co_{0.5}WO_4$; and (**d**) %degradation vs. catalyst dose graph.

### 2.3. Kinetics of Photodegradation

A variety of experiments were carried out by taking 10 mL of 20 ppm XO with 10 mg of catalyst at varying irradiation time from 0 to 120 min under visible solar radiation, and the data obtained was adjusted to the Langmuir–Hinshelwood (L–H) pseudo-first-order kinetic model for the quantitative analysis of XO degradation [37,38].

$$-\ln\left(\frac{C_e}{C_o}\right) = k_{app} \times t \qquad (6)$$

where $C_o$ represents the initial concentration and $C_t$ represents the final concentration of XO at time t. The results are given in Figure 10a–d, in which it was observed that with an increase in irradiation time, the photocatalytic efficiency increases. This may be due to an increase in irradiation time, the surface of the catalyst becomes more activated by absorbing the solar radiation and generates a greater number of ROS $^\bullet OH$ or $^\bullet O_2^-$ radicals which interact with dye molecules and degrade them [39]. The slope of the pseudo-first-order reaction (Figure 10d) can be utilized to determine the value of $k_{app}$ from the plot of $-\ln(C_e/C_0)$ vs. time (min), which shows a linear response. The obtained values of $k_{app}$ given in Table 2 as 0.018 $min^{-1}$ for $ZnWO_4$, 0.021 $min^{-1}$ for $CoWO_4$, and 0.025 $min^{-1}$ for $Zn_{0.5}Co_{0.5}WO_4$ proved the high efficiency of $Zn_{0.5}Co_{0.5}WO_4$ as compared to pristine $ZnWO_4$ and $CoWO_4$. The obtained value of $R^2 = 0.99$ for all the nanoparticles suggested that the photocatalytic data can be realistically simulated by the L–H model. The outcomes demonstrate that the deposition of $Co^{2+}$ in $ZnWO_4$ of both catalysts considerably improves

the degrading ability compared to each catalyst individually. Finally, $Zn_{0.5}Co_{0.5}WO_4$ was the novel composite having distinctive characteristics when compared to other composites.

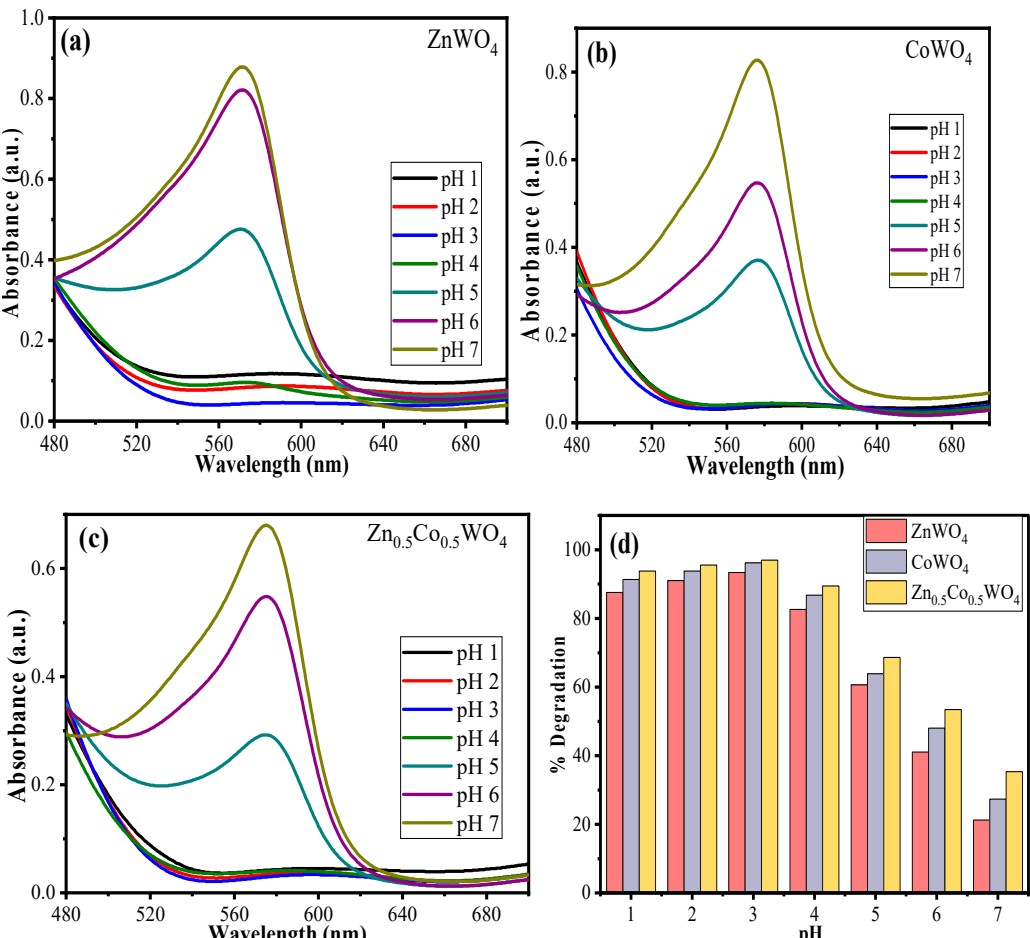

**Figure 9.** Effect of solution pH on photocatalytic degradation by (**a**) $ZnWO_4$; (**b**) $CoWO_4$; (**c**) $Zn_{0.5}Co_{0.5}WO_4$; (**d**) XO degradation (%) vs. pH bar graph.

**Table 2.** The kinetic parameters for the degradation of XO dye (20ppm) using 10 mg $ZnWO_4$, $CoWO_4$, and $Zn_{0.5}Co_{0.5}WO_4$ with variable irradiation time (5–120 min).

| Material | $k_1$ (min$^{-1}$) | $R^2$ | $t_{1/2}$ (min) |
|---|---|---|---|
| $ZnWO_4$ | 0.018 | 0.99 | 38.50 |
| $CoWO_4$ | 0.021 | 0.99 | 33.00 |
| $Zn_{0.5}Co_{0.5}WO_4$ | 0.025 | 0.99 | 27.72 |

## 2.4. Scavenging Study and Mechanism of Photodegradation

It was suggested that a variety of reactive species, including the hydroxyl radical ($^\bullet OH$), superoxide ($^\bullet O_2^-$), valence band hole (h$^+$), and electron (e$^-$) play important roles in the photodegradation of organic pollutants, and their effectiveness depends upon the band structure and chemical nature of the photocatalysts [40]. So, scavenger studies were performed using a 1.3 mM solution of Benzoic acid (BA), Potassium dichromate ($K_2Cr_2O_7$), (AA), ethylenediaminetetraacetic acid (EDTA), and benzoquinone (BQ) by taking 10 mL of 20ppm XO with 10 mg at pH 3 for an irradiation time of 120 min. We know that BA is a scavenger for $^\bullet OH$, EDTA for h$^+$, $K_2Cr_2O_7$ for e$^-$, and BQ for $^\bullet O_2^-$ [41,42]. Figure 11 displays the results, which show that without the use of a scavenger, the degrading efficiency of $Zn_{0.5}Co_{0.5}WO_4$ towards XO achieved 98.27%. After adding BA and EDTA, there appears no appreciable change in photocatalytic efficiency, suggesting no primary role of $^\bullet OH$

and $h^+$ on the degradation of XO at the optimized conditions. On the contrary, when BQ and $K_2Cr_2O_7$ are added, an appreciable decrease in photocatalytic efficiency happens, demonstrating that photogenerated electrons ($e^-$) and $^\bullet O_2{}^-$ radicals are predominant species in this photodegradation reaction scheme. The results showed that the degradation of XO was primarily controlled by $e^-$ and $^\bullet O_2{}^-$ radicals (majorly).

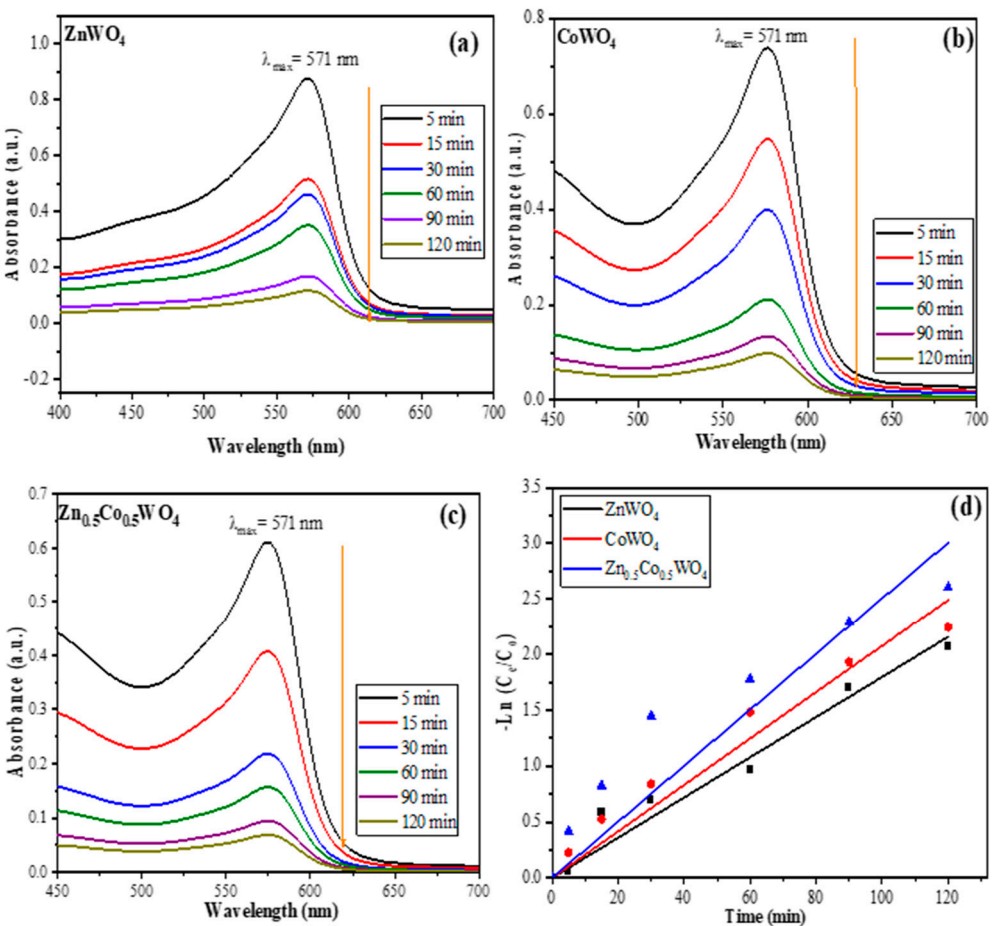

**Figure 10.** UV–vis absorption spectra of degraded XO dye (60 ppm) using 10 mg of (**a**) $ZnWO_4$; (**b**) $CoWO_4$; (**c**) $Zn_{0.5}Co_{0.5}WO_4$; and (**d**) L–H model pseudo-first-order kinetic graph with variable irradiation time (5–120 min).

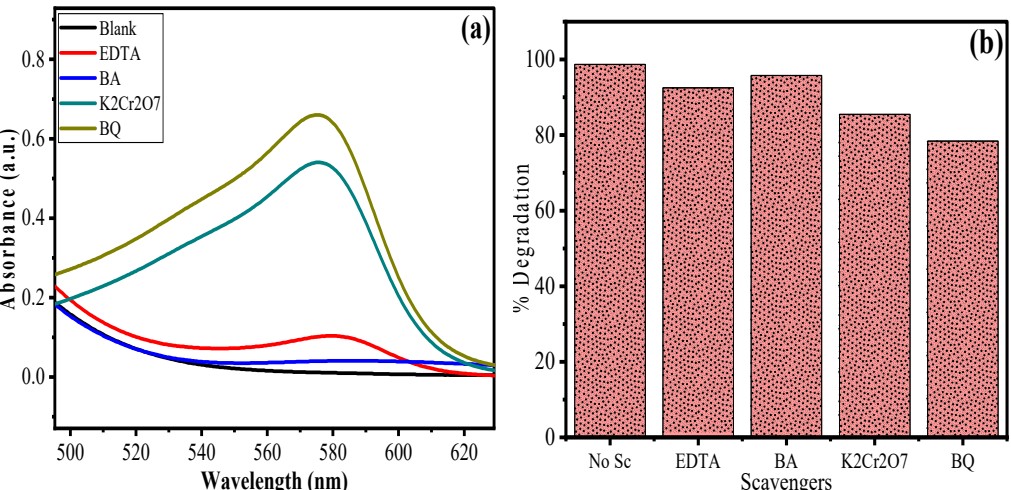

**Figure 11.** (**a**) UV–vis spectra of XO degradation in the presence of various scavengers; (**b**) % XO degradation vs. bar graph associated with each scavenger.

The plausible mechanism of degradation of XO by $\bullet O_2{}^-$ radicals is given in Figure 12 [43,44].

**Figure 12.** A plausible mechanism of degradation of XO by synthesized $Zn_{0.5}Co_{0.5}WO_4$.

*2.5. Comparison with the Literature*

The outcomes of this study were compared with the information available in the literature, and a comparison in Table 3 was made to investigate the upgraded scientific information added by the present study. It was found that there is no research where tungstate base nanomaterial is used for the photocatalytic degradation of xylenol orange.

**Table 3.** Comparison of the literature information with the present study.

| Catalysts | Irradiation Time (min) | Light Source | Dye Used | % Degradation | References |
|---|---|---|---|---|---|
| $BiOBr/ZnWO_4$ | 170 | UV-A light | RhB | 99.40 | [45] |
| La: $ZnWO_4$ | 90 | UV light | MB | 97.00 | [46] |
| $CoWO4/ZnWO_4$ p-n heterojunction | 40 | Xe lamp | RhB | 93% | [47] |
| $ZnWO4/WO_3$ heterojunction | 120 | Visible light | MB | 83.60 | [48] |
| $ZnWO4$-(ZnO) | 120 | UV-illumination | MO | 99.00 | [49] |
| CDs-$ZnWO_4$ | 150 | UV-illumination | NGB | 93.00 | [50] |
| $Zn_{0.5}Co_{0.5}WO_4$ | 120 | Visible Solar Light | XO | 98.77 | Present Study |

*2.6. Reusability Test*

The reusability test was performed for the synthesized $Zn_{0.5}Co_{0.5}WO_4$ up to five cycles to assess the stability of the material towards XO degradation, and the results are given in Figure 13a. It can be seen that there is no appreciable change in photocatalytic efficiency up to the first two cycles with greater than 98% of XO degradation. Furthermore, from cycle 3 to cycle 5, there appears some appreciable decrease in photocatalytic efficiency attaining 85% of XO degradation in cycle 5. The overall results suggest that the synthesized $Zn_{0.5}Co_{0.5}WO_4$ have good stability toward the XO degradation under optimized reaction conditions. To support the stability of the material, XRD analysis of the synthesized material before and after the photocatalytic reaction was taken into consideration, which is given in Figure 13b. The XRD spectra of $Zn_{0.5}Co_{0.5}WO_4$ after five consecutive cycles of reuse represented almost similar hkl planes, but with reduced intensity as compared to pristine $Zn_{0.5}Co_{0.5}WO_4$ before the photocatalytic reaction. Only one change was observed, that after the photocatalytic reaction the (-111) plane diminished, which may be due to the absorption of water.

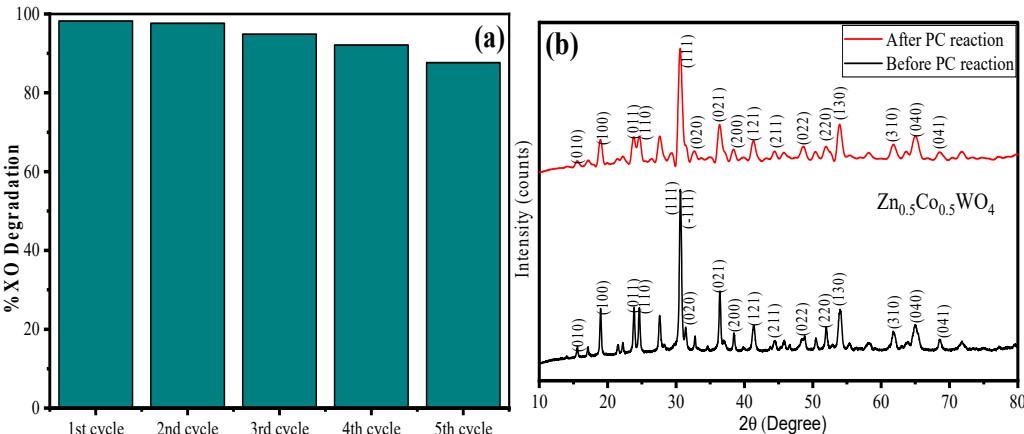

**Figure 13.** (**a**) Reusability test for synthesized $Zn_{0.5}Co_{0.5}WO_4$ NPs for XO degradation (**b**) XRD spectra of $Zn_{0.5}Co_{0.5}WO_4$ before and after the photocatalytic (PC) reaction.

## 3. Materials and Methods

### 3.1. Chemicals and Reagents

Sodium tungstate dihydrate ($Na_2WO_4 \cdot 2H_2O$, ACS grade, 99%) was purchased from Merck, Rahway, NJ, USA. Cobalt (II) nitrate hexahydrate (Co $(NO_3)_2.6H_2O$ ACS grade 98%), Zinc nitrate hexahydrate (Zn $(NO_3)_2.6H_2O$, reagent grade 98%), were purchased from Sigma Aldrich, St. Louis, Missouri USA. Xylenol orange ($C_{31}H_{28}N_2Na_4O_{13}S$, GR) and Ammonia solution (25%) were acquired from Otto Chemie, Mumbai, India. All the solutions were prepared in deionized (DI) water and used throughout all the photocatalytic experiments. Figure 14 represents the molecular structure of xylenol orange.

**Figure 14.** Structure of xylenol orange (XO).

### 3.2. Synthesis of $ZnWO_4$, $CoWO_4$, and $Zn_{0.5}Co_{0.5}WO_4$ Nanoparticles

The synthesis of pristine metal tungstate nanomaterials ($ZnWO_4$, $CoWO_4$ NPs) and mixed metal tungstate nanocomposite ($Zn_{0.5}Co_{0.5}WO_4$) was done according to the hydrothermal method reported elsewhere [51]. Three Erlenmeyer flasks of 50 mL capacity were equipped with 25 mL of 3mmol of sodium tungstate dihydrate ($Na_2WO_4 \cdot 2H_2O$), Zinc nitrate hexahydrate ($Zn(NO_3)_2 \cdot 6H_2O$; Flask 1), Cobalt(II)nitrate hexahydrate (Co $(NO_3)_2 \cdot 6H_2O$; Flask 2), and 1:1 ratio, i.e., 1.5 mmol each, of $Zn(NO_3)_2 \cdot 6H_2O$ and Co $(NO_3)_2 \cdot 6H_2O$ (Flask 3). All the flasks were placed on a magnetic stirring plate for 30 min to completely dissolve the salts and attain homogeneity. Then, the mixtures (F1, F2, and F3) were transferred into a Teflon-lined steel autoclave and heated in a convection oven at

180 °C for 24 h. The resulting precipitates of $ZnWO_4$, $CoWO_4$, and $Zn_{0.5}Co_{0.5}WO_4$ were centrifuged at 8000 rpm for 5 min and purified several times with DI water and finally with ethanol (2 times). After that, the materials were dried at 100 °C for 5 h and calcined at 600 °C for 4 h, then stored in a desiccator using a polystyrene Petri dish for further characterization and photocatalytic experiments.

### 3.3. Characterization Techniques

The $ZnWO_4$, $CoWO_4$, and $Zn_{0.5}Co_{0.5}WO_4$ were characterized using various techniques. The change in crystallite size and interplanar distance on Co deposition in the crystal lattice of $ZnWO_4$ was analyzed by an X-ray powder diffractometer (XRD, Rigaku Ultima IV, Tokyo, Japan) at Cu K$\alpha$ wavelength of 1.5418 Å as a source of radiation. The change of $WO_6^{6-}$ octahedron vibrations, Zn–O, and formation of Co–O bonding was assessed through Fourier transform infrared (FTIR) spectroscopy using a Perkin Elmer Spectrum 2 ATR spectrometer (Perkin Elmer, Waltham, MA, USA). The optical properties, energy bandgap of the synthesized NPs, and remaining concentration of XO in the effluent after the photocatalytic experiment was calculated by using a Shimadzu UV–1900 double-beam spectrophotometer (Kyoto, Japan). The surface morphology of the synthesized NPs was observed by scanning electron microscopy (SEM; JEOL GSM 6510LV, Tokyo, Japan) associated with an X-ray energy dispersive detector (EDX) to investigate the elemental composition and mapping. The morphological change in shape and size of the nanocrystalline solid upon Co deposition in $ZnWO_4$ was investigated using a transmission electron microscope (TEM; JEM 2100, Tokyo, Japan).

### 3.4. Photocatalysis Experiment

The photocatalytic efficiency of $ZnWO_4$, $CoWO_4$, and $Zn_{0.5}Co_{0.5}WO_4$ was analyzed towards the degradation of xylenol orange under visible light irradiation in a photocatalytic reactor. An aliquot of 10 mL of 50 ppm XO was taken in a 20 mL 1.8 cm glass tube with a magnetic bar of 10 mg of synthesized NPs in a photocatalytic reactor using Xe lamp (420 W cm$^{-1}$) for 5–120 min of irradiation. The aliquots are taken out of the photoreactor at a regular time interval, centrifuged to separate the nanocatalyst, and then placed in a UV–vis spectrophotometer at $\lambda_{max}$ = 580 nm to check the photocatalytic efficiency of the synthesized NPs, which is given by Equation (7);

$$XO\ Degradation(\%) = \left( \frac{C_0 - C_e}{C_0} \right) \times 100 \tag{7}$$

where $C_0$ and $C_e$ represents the initial concentration of an organic pollutant at time t = 0 and at any time t, respectively. The experiments were repeated by changing the parameters, for instance, organic pollutant concentration (ppm), irradiation time (min), catalytic load (mg), and pH of the media to optimize the reaction condition and predict the mechanism of the degradation.

### 4. Conclusions

The present study reveals the hydrothermal synthesis of pristine $ZnWO_4$, $CoWO_4$, and mixed metal $Zn_{0.5}Co_{0.5}WO_4$ nanoparticles (NPs) in a Teflon-lined autoclave at 180 °C for 24 h. The structural, morphological, elemental, and optical analyses supported the formation of $Zn_{0.5}Co_{0.5}WO_4$. The XRD analysis revealed a contraction in crystallite size from 32 nm to 24 nm with a decrease in crystallinity from 45% to 42% in $ZnWO_4$ to $Zn_{0.5}Co_{0.5}WO_4$ suggesting the successful deposition of $Co^{2+}$ in the $ZnWO_4$ lattice. The TEM analysis suggested the nanorod-like shape of the $Zn_{0.5}Co_{0.5}WO_4$ NPs with an average particle size of 31.74 nm. The optical studies revealed the UV–vis light activeness of materials with the energy band gap of $ZnWO_4$ (3.49 eV), $CoWO_4$ (3.42 eV), and mixed metal $Zn_{0.5}Co_{0.5}WO_4$ (3.33 eV). The material was explored as a catalyst for the photodegradation of XO dye under visible solar radiation and the results suggested the order of photocatalytic efficiency as $Zn_{0.5}Co_{0.5}WO_4$ (98.77%) > $CoWO_4$ (98.10%) > $ZnWO_4$ (97.89%) utilizing 10 mg

of catalyst for 60 ppm XO for 120 min of solar irradiation. The photocatalytic data were well explained by L–H pseudo–first–order kinetics with $R^2 = 0.99$, and apparent rate constant ($k_{app}$) as 0.018 min$^{-1}$ for ZnWO$_4$, 0.021 min$^{-1}$ for CoWO$_4$, and 0.025 min$^{-1}$ for Zn$_{0.5}$Co$_{0.5}$WO$_4$. The scavenger experiments disclosed that superoxide anion radicals (•O$_2$$^-$) and photogenerated electrons (e$^-$) played important roles in the degradation of XO. The outcomes of this study revealed that the eco-friendly and economical hydrothermal synthesized material (Zn0.5Co0.5WO4) can be used more efficiently for the treatment of XO-contaminated water in industries without the generation of any secondary pollution.

**Author Contributions:** Conceptualization, F.A.A.; Methodology, F.A.A., H.S.A. and I.H.; Software, I.H.; Validation I.H.; Investigation, W.S.A.-N. and A.A.A.; Resources, W.S.A.-N. and A.A.A.; Data curation, I.H.; Writing—original draft, W.S.A.-N.; Writing—review & editing, H.S.A. and I.H.; Visualization, I.H.; Supervision, F.A.A. and H.S.A.; Project administration, F.A.A.; Funding acquisition, F.A.A. All authors have read and agreed to the published version of the manuscript.

**Funding:** The authors extend their appreciation to the Deputyship of Research and Innovation, Ministry of Education in Saudi Arabia for funding this research work through project number (IFKSURG-2-255).

**Conflicts of Interest:** The authors declare no conflict of interest.

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
