# Peer review of "Hydrothermal Synthesis of Bimetallic (Zn, Co) Co-Doped Tungstate Nanocomposite with Direct Z-Scheme for Enhanced Photodegradation of Xylenol Orange"

_catalysts, doi:10.3390/catal13020404_

Round 1
Author Response
- It is claimed that ZnWO4, CoWO4 and ZnCoWO4 are synthesised in this article, but it seems there are many impurity peaks in xrd. Do you identify the impurity peaks at 2θ = ~17.2°, 21.6°, 22.2°, 27.6°,28.3°, 29.3°, 29.4°, 33.9° and 34.6° ? This must be an another phase and might be playing role in photocatalytic activity. In ZnWO4 xrd in Fig. 3, the main peak ~34.3° is splitted, you should mention both hkl planes (-111 is missing).
Thank you very much for considering our manuscript to review. In the XRD spectra of ZnCoWO4, we have just classified peaks belonging to ZnWO4 (black) and CoWO4 (purple) which suggest that in the single crystal lattice, both Zn and Co have occupied some of the lattice position at corners with WO66- frame in common. We have assigned the hkl plane for the split peak of ZnWO4.
- It is not able to identify shape of crystallites from SEM images, how you identify morphology from this kind of SEM data, its questionable. EDX also indicates impurity phase, as in ZnCoWO4, atomic % ratio of Zn: Co: W should be 0.33:0.33:0.33, but in the current EDX data there is an excessive amount of W, which indicates towards more than one phases.
The shape of the crystallite was finally confirmed by TEM analysis. The EDX data is correct according to our synthesis protocol in which we have taken 3 mmol Sodium tungstate with 1.5 mmol of both zinc nitrate and cobalt nitrate. It can be seen that the atomic ratio of Zn and Co is approximately 1:1.
- Why crystallinity is not reflected in TEM images, as there are no fringes observed in TEM images. HR-TEM is advisable to further proof its crystallinity and purity.
As per the reviewers’ suggestion, we have added the SAED image which shows the fringes belonging to the hkl planes of ZnCoWO4 with an extra TEM image recorded at 20 nm magnification.
- P-8, L-233; How you deduce that an increase in absorption and decrease in band gap can affect electron-hole recombination. Explain and cite some report.
An explanation with citation has been added to the manuscript.
- Fig. 7d is not mentioned in figure caption. Data in fig. 7d is not matching with the results of the experiment, as ZnWO4 showed least activity with 100 ppm XO, but according to Fig.7d it showed highest activity with 100 ppm XO, same for others also. Go, through your results and correct it in the Figure as well as text of the manuscript.
Fig.7d is now mentioned in the caption. It is basically a bar graph comparing the photocatalytic activity of ZnWO4 (yellow bar), CoWO4 (green bar) and ZnCoWO4 (purple bar) vs XO concentration which shows the maximum degradation of XO till 60 ppm concentration. On increasing the XO concentration beyond 60 ppm, photocatalytic efficiency (%) decreases.
- Section 3.4, First, why you choose these scavengers, generally EDTA, NaI, IPA/tert-BA and benzoquinone/ascorbic acid are used for hole, electron-, •OH and O2•⁻, respectively. How, you choose EDTA for •OH, acrylamide for hole, benzoic acid for electron and TPP for O2•⁻, they all wrong and the analysis based on these is also wrong as EDTA is used for holes, acrylamide for O2•⁻benzoic acid for •OH and TPP for electrons. Repeat the experiment and change the section 3.4
accordingly. See some references: doi.org/10.1016/j.jtice.2019.03.011;
doi.org/10.1021/acsomega.8b01054; doi.org/10.1007/s10854-019-00898-w
Thank you very much for the suggestion. We have repeated the experiment and cited the valuable references as R40, R41
- The generation of charged carriers and their separation should be demonstrated by some of the generally used techniques in photocatalysis, such as PL or EIS. It will be nice to prove photoexcited carrier generation (photocurrent and/or photovoltage, reference: DOI: 10.1039/D1CY01644JCatal. Sci. Technol., 2022, 12, 6704), their dynamics, lifetime and carrier separation, see reference:
DOI: 10.1039/D1MA00304F Mater. Adv., 2021, 2, 4832.
Thank you for the valuable suggestion. We have added PL spectra of the synthesized nanoparticles in the manuscript along with the references as R30, R31
- In abstract, P-1, L-26 it is difficult to understand this line, rewrite it.
The correction has been done in the abstract section accordingly.
- P-3, and L-35 it also needs to be rewritten.
The sentence has been revised.
- There are many typo errors in the manuscript, thus a thorough check on this manuscript is needed.
The authors have reviewed the whole manuscript and corrected all the typographical mistakes.
- Go through the references again, some errors in style and format.
The references have been revised according to the journal format.
Reviewer 2 Report
Manuscript ID: catalysts-2127219
Title: Hydrothermal Synthesis of Bimetallic (Zn, Co) Tungstate Na- nohetrojunction with Direct Z-Scheme for Enhanced Photodegradation of Xylenol Orange
Comments:
Major revision with these comments
1. Title is incorrect, word “nohetrojunction” need revision.
2. In introduction section, 1st sentence should be revised.
3. There are so many grammatical, spelling and formatting mistakes in manuscript. I can’t mentioned here all of it. Make sure the correction of all those and the whole manuscript must be cross-checked thoroughly for English editing, grammatical, spelling mistakes, and syntax errors.
4. In introduction section add latest literature on photocalysts and dyes degradation, suggested to cite, Optical Materials 126 (2022) 112199, Chemical Physics Letters 805 (2022) 139939, Journal of Molecular Liquids 356 (2022) 119036.
5. Section 2.1. there is error in font size and font style, please check and revise.
6. Effect of pH of dye solution should be examined and added in revised manuscript.
7. Fig.8 mentioned in caption sub figure a,b,c,d
8. Fig.9 mentioned in caption sub figure a,b,c,d also mark last sub Fig as d.
9. Reusability test should be performed to check the feasibility of application of commercial scale.
Author Response
- The title is incorrect, the word “nohetrojunction” need revision.
Thank you very much for considering our manuscript to review. The title has been revised now.
- In introduction section, 1stsentence should be revised.
The correction has been done accordingly.
- There are so many grammatical, spelling and formatting mistakes in manuscript. I can’t mentioned here all of it. Make sure the correction of all those and the whole manuscript must be cross-checked thoroughly for English editing, grammatical, spelling mistakes, and syntax errors.
The authors have gone through the whole manuscript and corrected all the typographical mistakes.
- In introduction section add latest literature on photocalysts and dyes degradation, suggested to cite, Optical Materials 126 (2022) 112199, Chemical Physics Letters 805 (2022) 139939, Journal of Molecular Liquids 356 (2022) 119036.
The recommended references were highly useful and have been cited in the manuscript as R7, R8, and R9.
- Section 2.1. there is an error in font size and font style, please check and revise.
Corrected now
- Effect of pH of dye solution should be examined and added in revised manuscript.
Experiments were performed for variable XO solution pH in the range of 1-7 and have been added in the manuscript now with the explanation.
- Fig. 8 mentioned in caption subfigure a,b,c,d
Corrected now
- Fig. 9 mentioned in caption sub figure a,b,c,d also mark last sub Fig as d.
Corrected now
- Reusability test should be performed to check the feasibility of application of commercial scale.
The reusability test has now been added to the revised manuscript.
Round 2
Reviewer 1 Report
Authors still need to answer the some of the major questions, and there are some new comments arise from new experiments.
1.Is the chemical formula for compound is Zn0.5Co0.5WO4 or is it a heterojunction of ZnWO4/CoWO4. I think, it is Zn0.5Co0.5WO4, as all XRD patterns well match and indexed in P2/C space gp. in monoclinic crystal system. Generally, heterojunction are synthesised in a two step method where different catalysts does not loose their identity, which can be verified by XRD or two different morphology in SEM (which is not clear). So, I stongly believe, it is Zn and Co co-doped compound instead of hetrojunction ZnWO4/CoWO4. So, remove heterojunction word from the manuscript and change the formula to Zn0.5Co0.5WO4 instead of ZnCoWO4 in the manuscript and in all the Figures. Change the Fig. 11 and discuss the mechanism in text also.
2.I think the peaks at 2θ = ~17.2°, 21.6°, 22.2°, 27.6°,28.3°, 29.3°, 29.4°, 33.9° and 34.6° does not belong to either of ZnWO4 and CoWO4. There must be an impurity, as your EDX is also showing higher content of W than expected.
3.In the EDX according to authors, Co :Zn should be 1:1, it is ok. But what about W? It should be double of Zn/Co. In the results Zn:Co:W is 1:1:5, iwhich is not acceptable, as according to authors it should be 1:1:2, which is almost 2.5 times higher.
4. In Fig. 12 it is hard to recognize any morphology from the data. So, it is advised to show the stability of catalyst by XRD data before and after photocatalyis.
Author Response
1.Is the chemical formula for compound is Zn0.5Co0.5WO4 or is it a heterojunction of ZnWO4/CoWO4. I think, it is Zn0.5Co0.5WO4, as all XRD patterns well match and indexed in P2/C space gp. in monoclinic crystal system. Generally, heterojunction are synthesised in a two step method where different catalysts does not loose their identity, which can be verified by XRD or two different morphology in SEM (which is not clear). So, I stongly believe, it is Zn and Co co-doped compound instead of hetrojunction ZnWO4/CoWO4. So, remove heterojunction word from the manuscript and change the formula to Zn0.5Co0.5WO4 instead of ZnCoWO4 in the manuscript and in all the Figures. Change the Fig. 11 and discuss the mechanism in text also.
Thank you for the suggestion. We have changed the formula and removed heterojunction from the manuscript accordingly.
2.I think the peaks at 2θ = ~17.2°, 21.6°, 22.2°, 27.6°,28.3°, 29.3°, 29.4°, 33.9° and 34.6° does not belong to either of ZnWO4 and CoWO4. There must be an impurity, as your EDX is also showing higher content of W than expected.
Yes, we agree with you. There are several peaks that do not belong to either ZnWO4 or CoWO4. These peaks are due to the formation of Na2W2O7 and Na2W4O13 phases with the Zn0.5Co0.5WO4 phase. Due to the formation of these phases, we are getting a high amount of W in the EDX spectra.
3.In the EDX according to authors, Co :Zn should be 1:1, it is ok. But what about W? It should be double of Zn/Co. In the results Zn:Co:W is 1:1:5, iwhich is not acceptable, as according to authors it should be 1:1:2, which is almost 2.5 times higher.
Thank you for your careful observation. The reason for the high amount of W in the EDX spectra has been explained in the previous comment with proper citation.
- In Fig. 12 it is hard to recognize any morphology from the data. So, it is advised to show the stability of catalyst by XRD data before and after photocatalyis.
We have substituted the SEM images with XRD analysis before and after photocatalysis as per the suggestion of the reviewer.
Reviewer 2 Report
Accept
Author Response
Thank you very much for reviewing and considering our manuscript for publication.